# Analysis of Factors Influencing the Corporate Performance of Listed Companies in China's Agriculture and Forestry Sector Based on a Panel Threshold Model

**Yong Sun [1,†], Hui Liu [2,†], Jiwei Liu [3,\*], Mingyu Sun [3] and Qun Li [4]**

1. Institute of Quantitative and Technological Economics, Chinese Academy of Social Sciences, Beijing 100732, China
2. School of Economics and Management, Nanjing Forestry University, Nanjing 210037, China
3. School of Applied Economics, University of Chinese Academy of Social Sciences, Beijing 102488, China
4. Institute of Ecological Development, China ECO Development Association, Beijing 100013, China
* Correspondence: liujiwei@ucass.edu.cn
† These authors contributed equally to this work.

**Abstract:** The global food crisis caused by COVID-19 and the Russia–Ukraine conflict have made many countries around the world realize the significance of agroforestry to a country's food security. However, China's agroforestry R&D innovation is currently lagging behind in development, and some agricultural seeds are heavily dependent on foreign countries, which seriously affects China's national food security. It is especially important to explore the reasons why China's agroforestry R&D and innovation is lagging behind. As listed agroforestry companies face the market demand directly, there is an urgent need to study the R&D innovations of listed agroforestry companies at present. This paper analyzes the impacts of R&D innovation, corporate management and supply chain management on the corporate performance of listed agroforestry companies using the entropy weighting method, GMM estimation and panel threshold model, mainly by selecting annual panel data from CSMAR for the period 2010 to 2021. The following conclusions were drawn: (1) There is a nonlinear relationship between R&D innovation and firm performance, and a "U"-shaped relationship. This indicates that there is an entrance threshold for R&D innovation in the agroforestry industry, below which corporate performance does not improve. (2) There is a nonlinear relationship between corporate management and corporate performance, and a U-shaped relationship. (3) There is a nonlinear relationship between supply chain management and firm performance, with an inverted-U-shaped relationship. This paper explains the reasons for the slow development of R&D innovation in China's agriculture and forestry industry and fills the gap in the theoretical study of the nonlinear relationship between R&D innovation and corporate performance of listed companies in China's agriculture and forestry industry. Finally, this paper provides a theoretical basis for the decision making of government departments related to agriculture and forestry, and offers some suggestions for listed companies in agriculture and forestry to improve their corporate performance.

**Keywords:** corporate performance; R&D innovation; supply chain management; Chinese agriculture and forestry; panel threshold model

## 1. Introduction

China is striving to reach peak $CO_2$ emissions by 2030 and to become carbon neutral by 2060 [1–3]. The global food and energy crises caused by the COVID-19 pandemic and the Russian–Ukrainian conflict have made the world realize the importance of complex agroforestry development for a country [4]. Both droughts and epidemics are disrupting global food supply chains, and the impact of the COVID-19 outbreak is likely to intensify, leading to severe economic stress and malnutrition, particularly in developing countries [5]. Farmers have adopted appropriate cropping systems in China according to climatic, soil,

and water conditions [6]. The government and the markets are the two main instruments for allocating resources, and both play an essential role in economic development and environmental protection [7]. There is growing awareness of the need to use natural resources sustainably and shift to a resource-efficient economy [8].

A good example is a joint program to address the challenges of COVID-19, conflict, and climate change which benefits the six countries of the Sahel by investing in agricultural infrastructure, innovative technologies, and human and social capital in cross-border areas facing security issues [9–11]. There is general awareness that as the land boundaries for further agricultural expansion shrink, future agricultural growth will increasingly have to come from increased productivity and resource use efficiency rather than from area expansion. Therefore, protecting and enhancing the innovation systems of the natural-resource-based industries while increasing productivity is a fundamental requirement for sustainable development [12,13]. However, agroforestry development in China has stagnated scientific and technological innovation in agroforestry development. Therefore, examining the scientific and technological innovations affecting agroforestry development in China is necessary.

The research value of this paper is mainly twofold: (1) In theory, our research fills the gap in the theoretical study of the nonlinear relationship between R&D innovation and corporate performance in agroforestry listed companies. (2) In practice, our research explains the reasons for the slow development of R&D innovation in the Chinese agriculture and forestry industry. It provides theoretical guidance for some of the relevant governmental decisions, and also gives some advice for listed agroforestry companies to improve their corporate performance by increasing R&D innovation.

In the face of the world's current challenges with maintaining good living conditions, including increasing population size, climate change, and the degradation of agroecosystems associated with declining agricultural productivity, approaches are needed to ensure food security. China's seeds industry lacks innovation, and soybean seeds highly depend on imports. Agricultural innovation, then, needs to be given high priority by society. As listed companies in agriculture and forestry directly face the market demands and are more able to grasp the market's direction of agricultural products, it has become imperative to study the research innovation of listed companies in agriculture and forestry. We selected annual panel data of listed companies in the agriculture and forestry category from the CSMAR database for the period 2010–2021. A panel threshold model was used to explore the nonlinear relationship between R&D innovation, corporate governance, supply chain management, and corporate performance of listed agroforestry companies in five dimensions. At the same time, the reasons for the lagging R&D innovation in China's agroforestry industry were investigated in depth using GMM estimation methods, theoretically filling a gap in the field of research on the nonlinear relationship between R&D innovation and corporate performance of listed companies in the agroforestry industry.

The rest of the paper proceeds as follows. Section 2 is the literature review section, Section 3 is the data processing section, Section 4 is the empirical analysis section, and Section 5 is the conclusion and recommendations section. Section 6 is the discussion section.

## 2. Literature Review

In the background of ecological civilization and food security, environmental, climate, and food issues have gained much attention and become some of the hot global research issues. The problem of global warming is becoming increasingly severe. The worldwide average atmospheric $CO_2$ concentration reached a peak of 421 ppm in May 2022, another record high [14]. Global demand for food continues to increase as the population grows, but the limited and scarce natural resources needed to produce food are accompanied by ecological degradation and crisis [15]. In response, the Chinese government has introduced several environmental projects in degraded areas over the past two decades, involving farmers in creating economically and environmentally sound technologies that sustainably and equitably manage natural resources [16,17]. With increasing population, climate change,

and other changes in the external environment, natural resources, including land, are becoming increasingly scarce, and ecological problems are becoming more pronounced.

Agroforestry development is an excellent solution to the conflict between conservation and development and environmental and food problems. Many scholars have researched this area and made practical recommendations for agroforestry development. Agroforestry development is closely related to the climatic environment, and its development has a strongly positive ecological and environmental effect [18]. Studies have concluded that the carbon sequestration capacity of agroforestry complex systems is significant [19]. As a result, the IPCC recommended the agroforestry system as a land-use model for sequestering atmospheric $CO_2$. The development of agroforestry will increase the value of other agricultural and forest resources, thereby alleviating ecological and developmental conflicts, which in turn will lead to local development [20]. The structural complexity of agroforestry is an important driver of the diversity of ecological, economic and resource functions of agroforestry systems [21]. Agroforestry development can be pursued in three ways. First, to effectively promote agroforestry, relevant measures must integrate ecological and economic aspects [22]. Secondly, the scientific identification and mitigation of conflicts between agricultural and environmental functions is the key to managing national land resources and optimizing the spatial pattern of the land [23]. Thirdly, a trade-off should be made between the operational efficiency of the agricultural management system and the system's integrity to achieve an optimal balance according to the needs of ecological management and protection [24].

Scholars generally agree that R&D innovation in agroforestry is an effective measure to accelerate the transformation and optimization of agroforestry development, as a solution to the significant problem of resource and environmental constraints. However, there is a lack of nonlinear theoretical research on R&D innovation in listed agroforestry companies, so this paper takes the nonlinear relationship between R&D innovation and the enterprise performance of listed agroforestry companies as the landing point to carry out research. In terms of economic and social development, agroforestry enterprises need innovation. Particularly in the core competencies of firms, the translation of agroforestry science and technology innovations into agroforestry technologies that benefit all stakeholders improves social and economic conditions in developing countries [25]. Climate-smart agriculture (CSA) contributes to increased productivity and food security in terms of food security, agricultural development, and climate improvement. CSA also helps to increase the resilience of farming systems, reduce greenhouse gas emissions, and sequester carbon [26]. In terms of ecological governance, green transformation, and production efficiency, accelerating the green change of agriculture through agricultural science and technology innovation is an effective measure to reduce farm pollution and improve agricultural production efficiency in the face of increasingly severe resource and environmental constraints [27]. Among them, green innovation is essential for agroforestry to achieve sustainable development and green transformation [28]. There are spillover effects in agroforestry innovation development, and factors that influence the process of innovation diffusion and dissemination in regional agroforestry development include physical carrying capacity, farmers' characteristics, facilities and infrastructure, accessibility, institutions, capital ownership, and partnerships [29]. Therefore, in-depth research can be conducted on the issue of innovation in listed agroforestry companies.

Supply chain management is an essential factor affecting the development of agroforestry enterprises, and in particular, the agroforestry product supply chain (ASC) has received increasing attention in recent years [30]. Scholars on supply chain management's importance, problems and improvement measures have conducted a great deal of research. Supply chain management in agroforestry enterprises is more complex than typical manufacturing supply chain management [31]. Agroforestry companies need to change their traditional supply chains and move towards sustainable ones that create products and services with sustainable thinking and ideas [32]. Specific measures include, firstly,

supply chain collaboration [33]. The second is the integrated planning of supply chain activities [34]. The third is the direct farm supply model [35].

Corporate management is an essential factor affecting the development of agroforestry firms and can directly influence firm performance [36]. The growth capacity and competitiveness of listed agroforestry companies can affect performance [37]. On the other hand, solvency indirectly affects agroforestry firms' performance levels by influencing trade of agricultural products [38]. Agricultural policies and management systems can also affect the performance of agricultural firms and business development [39].

In summary, scholars have focused their research on agroforestry development in these areas of ecological governance, environmental protection, climate improvement, and food security. However, most of these studies are based on linear relationships, and there are relatively few studies on nonlinear relationships. Currently, China's central bodies of technological innovation in agriculture and forestry are mainly government research institutions and agricultural and forestry colleges. Administrative features, serious bureaucratization, and redundant institutions typically characterize these institutions. However, we believe that the main body of agroforestry technology innovation in China should be agroforestry enterprises. This is because agroforestry enterprises are directly exposed to market demand, and only these can quickly transform innovative technologies into productivity; market demand guides innovation in agroforestry science and technology. At present, the promotion of new agricultural and forestry technology in China is not optimistic; the promotion system in many areas exists in name only, and small farmers do not have access to technical information and guidance. Listed agroforestry companies are essential and typical representatives of agroforestry enterprises, leading the direction of agroforestry technology innovation development. However, existing research is inadequate. Therefore, based on the current situation of lagging R&D innovation in China's agroforestry industry, we have raised the following three questions.

Question 1: What factors may affect R&D innovation in developing listed agroforestry companies?

Question 2: What factors may affect company management in developing listed agroforestry companies?

Question 3: What factors may affect supply chain management in developing listed agroforestry companies?

## 3. Variable Selection and Hypothesis Formulation

### 3.1. Data Sources

The data in this paper were sourced from the China Stock Market and Accounting Research Database (CSMAR), the National Bureau of Statistics, the State Forestry Administration, the Oriental Wealth Network, etc. The annual data of 40 A-share listed companies in the agriculture, forestry, and fishery industries were selected from 2010 to 2021. Since there are missing values in some years, we used the mean interpolation method and the nearest neighbor interpolation method to fill in the missing values. Then, we normalized the data and then reduced dimensionality using the entropy weighting method.

### 3.2. Variable Selection

#### 3.2.1. Explanatory Variables

In this paper, six primary indicators were selected, including the explanatory variable of corporate performance (CP), the explanatory variables of research and development (R&D), corporate management (CM), and supply chain management (SCM), and the control variables of growth capacity (Growth) and debt service capacity (DSC). In this paper, 32 secondary indicators were selected, including financial and non-financial ones.

(1) Research and development innovation (R&D)

A total of seven secondary indicators were selected for the explanatory variable R&D innovation, including the number of R&D personnel [40], the proportion of personnel in R&D (%), the amount of R&D investment [41], the proportion of R&D investment of

operating revenue (%), the amount of R&D investment (expenditure) labeled as expenses, the amount of R&D investment (expenditure) generating capital [42], and the proportion of capital converted to R&D investment (expenditure) (%). Some scholars have argued that the amount of R&D investment and the number of R&D personnel significantly impact firms' technological innovation performance in China's high-level economic environment [43,44].

(2) Corporate management (CM)

A total of nine secondary indicators were selected for the explanatory variable corporate management, including equity concentration, indicator1 (%); the board size [45], whether the effective controller is the chairman or general manager; number of shares held by the chairman [46]; percentage of shares held by the chairman (%); total remuneration of the top three executives; total remuneration of executives [47]; number of executives; and number of shares held by executives [48].Some scholars examined the relationship between executive compensation and corporate performance using indicators such as whether the beneficial owner is the chairman or managing director, total executive compensation, number of executives, and number of executive shares. Boards were expected to have more power, CEOs received less total cash and total compensation, and influential directors also appeared to establish a more vital link between CEO pay and corporate performance [49].

(3) Supply Chain Management (SCM)

A total of five secondary indicators were selected for the explanatory variable firm management, including net inventory [50], accounts payable turnover [51], total asset turnover [52], accounts receivable turnover [53], and inventory turnover. Supply chain management in agribusiness was studied using indicators such as total assets and inventory turnover [54].

### 3.2.2. Explanatory Variables

The explanatory variable corporate performance (CP) was selected from four secondary indicators: return on net assets [55], return on investment [56], operating profit margin [57], and return on total assets [58]. Financial indicators such as return on net assets, return on total assets and earnings per share play an essential role in predicting profitability [59].

### 3.2.3. Control Variables

(1) Growth capacity (Growth)

The control variable, growth capacity (Growth), was selected from four secondary indicators, including the growth rate of return on net assets [60], the growth rate of net profit [61], the growth rate of operating income [62], and the growth rate of net assets per share. Reference [63] studied the growth capacity of enterprises in the market-risk early warning model using indicators such as net profit growth rate and operating income growth rate.

(2) Debt Service Capacity (DSC)

A total of three secondary indicators were selected for the control variable debt-servicing capacity (DSC), including cash ratio [64], equity ratio [65], and gearing ratio [66]. The gearing ratio reflects the company's capital structure, and the gearing ratio can be a good indicator of the company's debt problem [67].

### 3.3. Hypothesis Formulation

As shown in Figure 1, the 3D mapping of R&D innovation, firm management, and corporate performance shows no simple linear relationship between R&D innovation, corporate management, and corporate performance. There is a "U"-shaped relationship between R&D innovation and corporate performance and a "U"-shaped relationship between corporate management and corporate performance.

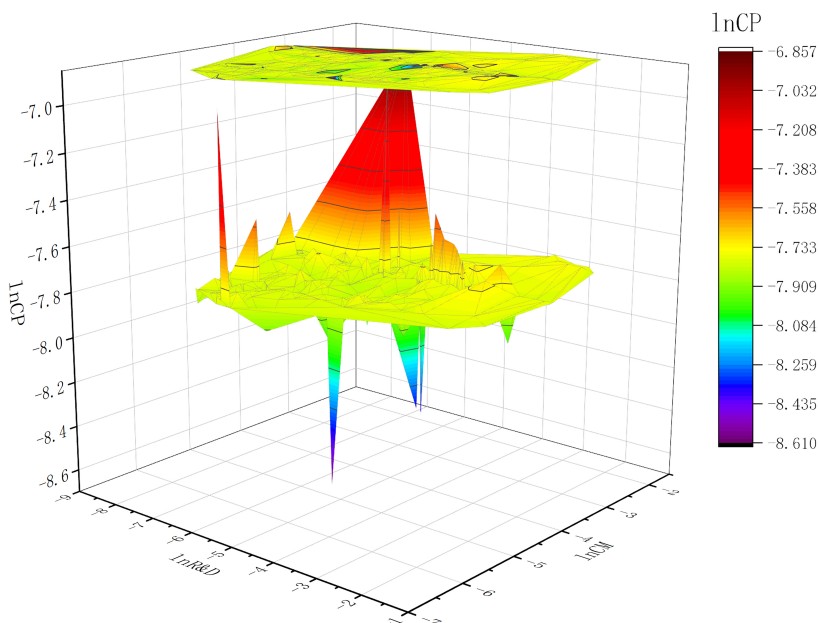

**Figure 1.** 3D diagram of lnR&D, lnCM, and lnCP.

As shown in Figure 2, the 3D mapping of R&D innovation, supply chain management, and corporate performance shows that the relationship between R&D innovation, supply chain management, and corporate performance is not linear. The relationship between R&D innovation and corporate performance is U-shaped, and the relationship between supply chain management and corporate performance is inverted-U-shaped.

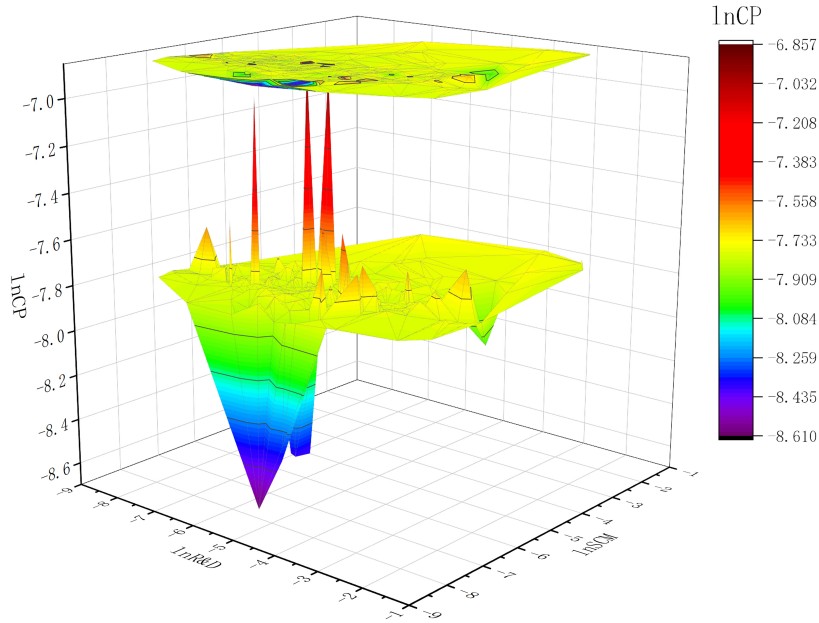

**Figure 2.** 3D diagram of lnR&D, lnSCM, and lnCP.

As shown in Figure 3, the 3D mapping of corporate management, supply chain management, and corporate performance shows that the relationships among corporate management, supply chain management, and corporate performance is linear. The relationship between corporate management and corporate performance is U-shaped, and the relationship between supply chain management and corporate performance is inverted-U-shaped.

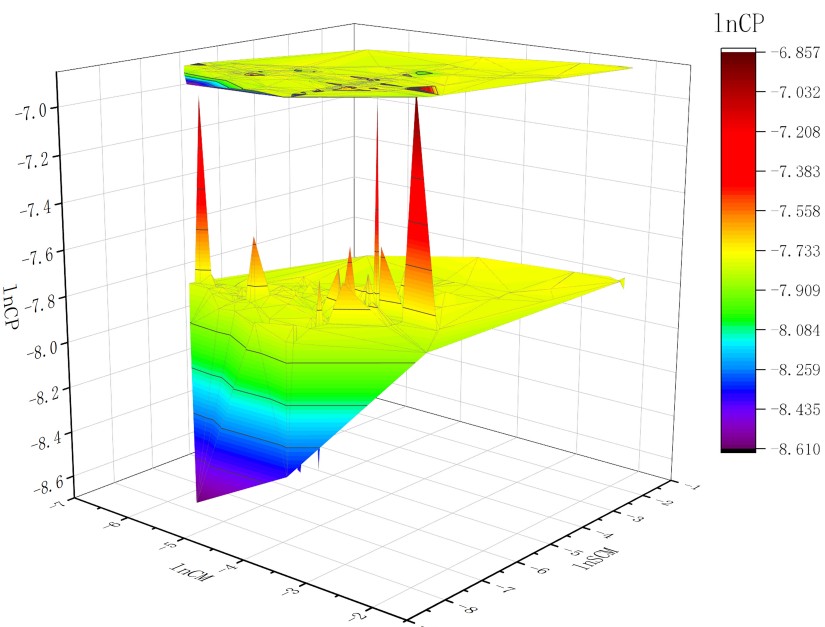

**Figure 3.** 3D diagram of lnCM, lnSCM, and lnCP.

In order to further explore the relationship between R&D innovation, company management, and supply chain management in the development of listed agroforestry companies, the following three hypotheses were formulated. Since the 1980s, scholars have studied the nonlinear relationship between R&D innovation and firm performance extensively [68,69]. Therefore, this paper proposes Hypothesis 1.

**Hypothesis 1.** *There may be a non-linear and U-shaped relationship between R&D innovation and corporate performance.*

Most theoretical studies on the relationship between corporate management and firm performance were linear relationship studies, using OLS, FE, 2SLS, and other models [70,71]. However, there are some scholars who argue for a U-shaped relationship between corporate governance and corporate performance [72–74]. Therefore, this paper proposes Hypothesis 2.

**Hypothesis 2.** *There may be a non-linear and U-shaped relationship between corporate management and corporate performance.*

In addition, some scholars have suggested that the relationship between supply chain management and firm performance may have an inverted-U-shaped relationship [75,76]. Therefore, Hypothesis 3 is proposed in this paper.

**Hypothesis 3.** *There may be a non-linear and inverted-U-shaped relationship between supply chain management and corporate performance.*

*3.4. Entropy Weighting Method*

We used the entropy weighting method to reduce the dimensionality of the data and determine the indicator weights. The entropy weighting method is a relatively objective method of assigning weights. The entropy method is a relatively objective method of assigning weights to indicators using the amount of information provided by the entropy value of each indicator. The detailed calculation process of the entropy method is shown in Appendix A. Finally, through the calculation of the entropy method, we obtained the results of the construction of the indicator system and the weights for indicators in this paper, as shown in Table 1.

**Table 1.** Results of the construction of the indicator system and the assignment of indicator weights.

| Variables | Tier 1 Indicators | Secondary Indicators | Weights |
|---|---|---|---|
| Explanatory variables | Research and Development Innovation (R&D) | X1 = Number of R&D staff | 0.025348 |
| | | X2 = Number of R&D staff as a percentage (%) | 0.015784 |
| | | X3 = Amount of R&D investment | 0.033095 |
| | | X4 = R&D investment as a percentage of operating revenue (%) | 0.023971 |
| | | X5 = Amount of R&D inputs (expenses) expensed | 0.033869 |
| | | X6 = Amount of R&D investment (expenditure) capitalized | 0.076709 |
| | | X7 = Capitalized R&D investment (expenditure) as a percentage of R&D investment (%) | 0.051822 |
| | Corporate Management (CM) | X8 = Equity concentration indicator1 (%) | 0.003983 |
| | | X9 = Board size | 0.005340 |
| | | X10 = Whether the actual controller is the chairman or general manager | 0.018003 |
| | | X11 = number of shares held by the chairman | 0.046478 |
| | | X12 = Chairman's shareholding (%) | 0.081623 |
| | | X13 = Total compensation of top three executives | 0.037186 |
| | | X14 = Total executive compensation | 0.074338 |
| | | X15 = Number of executives | 0.001475 |
| | | X16 = number of shares held by executives | 0.050831 |
| | Supply Chain Management (SCM) | X17 = Net Inventory | 0.021952 |
| | | X18 = Accounts payable turnover ratio | 0.095599 |
| | | X19 = Total asset turnover ratio | 0.033429 |
| | | X20 = Accounts receivable turnover ratio | 0.062244 |
| | | X21 = Inventory turnover ratio | 0.051612 |
| Control variables | Growth capacity (Growth) | X22 = Revenue on net assets growth rate | 0.012597 |
| | | X23 = Net profit growth rate | 0.000108 |
| | | X24 = Operating income growth rate | 0.121145 |
| | | X25 = Net asset per share growth rate | 0.000072 |
| | Debt Service Capacity (DSC) | X26 = Cash ratio | 0.013378 |
| | | X27 = Equity ratio | 0.003005 |
| | | X28 = Gearing ratio | 0.003859 |
| Explained variables | Corporate Performance (CP) | X29 = Revenue on net assets | 0.000133 |
| | | X30 = Revenue on investment | 0.000860 |
| | | X31 = operating profit margin | 0.000059 |
| | | X32 = Revenue on total assets | 0.000090 |

## 4. Empirical Analysis

### 4.1. GMM Model

4.1.1. Model Construction

(1) To test Hypothesis 1: there may be a non-linear and U-shaped relationship between R&D innovation and corporate performance. We used a GMM model to test the possibility of nonlinearity. Therefore, Model 1 is shown in Equation (7).

$$CP_{it} = \alpha_i + \beta_1 \cdot \ln R\&D_{it} + \beta_2 \cdot \ln R\&D^2_{it} + \sum \lambda_i \cdot \text{Control}_{it} + \varepsilon_{it} \tag{1}$$

(2) To test Hypothesis 2: There may be a non-linear and U-shaped relationship between corporate governance and corporate performance. We used a GMM estimation model to test the possibility of nonlinearity. Therefore, Model 2 is shown in Equation (8).

$$CP_{it} = \alpha_i + \beta_1 \cdot \ln CM_{it} + \beta_2 \cdot \ln CM^2_{it} + \sum \lambda_i \cdot \text{Control}_{it} + \varepsilon_{it} \tag{2}$$

(3) To test Hypothesis 3: There may be a non-linear and inverted-U-shaped relationship between supply chain management and corporate performance. We used a GMM estimation model to test the possibility of nonlinearity. Therefore, Model 3 is shown in Equation (9).

$$CP_{it} = \alpha_i + \beta_1 \cdot \ln SCM_{it} + \beta_2 \cdot \ln SCM^2_{it} + \sum \lambda_i \cdot \text{Control}_{it} + \varepsilon_{it} \qquad (3)$$

### 4.1.2. Results of GMM Estimation

The results of the GMM estimation are shown in Table 2. Table 2 shows that the primary and secondary coefficients of R&D innovation in Model 1 are positive, and the p-values are significant at the 1% level. Therefore, we can speculate on the possibility that Hypothesis 1 exists; i.e., there may be a non-linear relationship between R&D innovation and corporate performance and a U-shaped relationship. Table 2 also shows that the primary and secondary coefficients of Model 2 are positive, and the p-values are significant at the 5% level. Therefore, we can speculate that Hypothesis 2 is true—i.e., that there may be a non-linear relationship between corporate management and corporate performance—a U-shaped one. We can also conclude from Table 2 that both the primary and secondary coefficients of supply chain management in Model 3 are harmful, and the p-values are significant at the 5% and 1% levels, respectively. Therefore, we further speculate on the possibility that Hypothesis 3 is true—i.e., there may be a non-linear relationship between supply chain management and corporate performance—an inverted-U-shaped one.

**Table 2.** Results of GMM estimation.

| Variables | CP | | |
| --- | --- | --- | --- |
| | **Model 1** | **Model 2** | **Model 3** |
| *lnR&D* | 0.0000732 *** | | |
| | (−0.0000243) | | |
| *lnR&D*$^2$ | 0.00000693 *** | | |
| | (-0.0000023) | | |
| lnCM | | 0.0000548 ** | |
| | | (−0.0000226) | |
| lnCM2 | | 0.00000637 ** | |
| | | (−0.00000292) | |
| lnSCM | | | −0.0000223 ** |
| | | | (−0.00000946) |
| lnSCM2 | | | −0.00000279 *** |
| | | | (−0.00000107) |
| D_DSC | −0.00252 * | −0.00254 ** | −0.00265 * |
| | (−0.00147) | (−0.00128) | (−0.00147) |
| D_SCM | 0.000420 *** | 0.000306 *** | |
| | (−0.00015) | (−0.0000908) | |
| D_CM | | | 0.000210 * |
| | | | (−0.000123) |
| Constant | 0.000595 *** | 0.000528 *** | 0.000387 *** |
| | (−0.0000592) | (−0.0000419) | (−0.0000162) |
| $R^2$ | 0.036 | 0.008 | 0.036 |
| Observations | 440 | 440 | 440 |

Note: *** indicates significance at the 1% level; ** indicates significance at the 5% level; * indicates significance at the 10% level.

We used the GMM estimation model only to test whether the quadratic coefficients of the explanatory variables R&D innovation (R&D), corporate management (CM), and supply chain management (SCM) are significant as a way of supporting the possibility of the existence of a non-linear relationship between them. From the results of the above tests, we can roughly speculate on the likelihood of the three hypotheses being valid. However, we need more results.

### 4.2. Panel Threshold Models

4.2.1. Model Construction

To test Hypothesis 1: There may be a non-linear relationship between R&D innovation and corporate performance. We built the following four models using corporate management (D_CM), supply chain management (lnSCM), growth capability (D_Growth), and solvency (D_DSC) as threshold variables, respectively. Models 1–4 are shown in Equations (10)–(13).

$$\ln CP_{it} = \mu_i + \xi_1 \cdot \ln R\&D_{it} \cdot I \cdot (D\_CM_{it} \leq \gamma_1) + \xi_2 \ln R\&D_{it} \cdot I \cdot \\ (\gamma_1 < D\_CM_{it} \leq \gamma_2) + \xi_3 \cdot \ln R\&D_{it} \cdot I \cdot (D\_CM_{it} > \gamma_2) \\ + \sum \eta_i \cdot Control_{it} + \varepsilon_{it} \tag{4}$$

$$\ln CP_{it} = \mu_i + \xi_1 \cdot \ln R\&D_{it} \cdot I \cdot (\ln SCM_{it} \leq \gamma_1) + \xi_2 \cdot \ln R\&D_{it} \cdot I \cdot \\ (\gamma_1 < \ln SCM_{it} \leq \gamma_2) + \xi_3 \cdot \ln R\&D_{it} \cdot I \cdot (\ln SCM_{it} > \gamma_2) \\ + \sum \eta_i \cdot Control_{it} + \varepsilon_{it} \tag{5}$$

$$\ln CP_{it} = \mu_i + \xi_1 \cdot \ln R\&D_{it} \cdot I \cdot (D\_Growth_{it} \leq \gamma_1) + \xi_2 \cdot \ln R\&D_{it} \cdot I \cdot \\ (\gamma_1 < D\_Growth h_{it} \leq \gamma_2) + \xi3 \cdot \ln R\&D_{it} \cdot I \cdot (D\_Growth_{it} > \gamma_2) \\ + \sum \eta_i \cdot Control_{it} + \varepsilon_{it} \tag{6}$$

$$\ln CP_{it} = \mu_i + \xi_1 \cdot \ln R\&D_{it} \cdot I \cdot (D\_DSC_{it} \leq \gamma_1) + \xi_2 \cdot \ln R\&D_{it} \cdot I \cdot \\ (\gamma_1 < D\_DSC_{it} \leq \gamma_2) + \xi_3 \cdot \ln R\&D_{it} \cdot I \cdot (D\_DSC_{it} > \gamma_2) \\ + \sum \eta_i \cdot Control_{it} + \varepsilon_{it} \tag{7}$$

To test Hypothesis 2: There may be a non-linear relationship between corporate management and corporate performance. We developed the following three models using supply chain management (lnSCM), growth capacity (D_Growth), and solvency (D_DSC) as threshold variables, respectively. Models 5–7 are shown in Equations (14)–(16).

$$\ln CP_{it} = \mu_i + \xi_1 \cdot \ln CM_{it} \cdot I \cdot (SCM_{it} \leq \gamma_1) + \xi_2 \cdot \ln CM_{it} \cdot I \cdot \\ (\gamma_1 < SCM_{it} \leq \gamma_2) + \xi_3 \cdot \ln CM_{it} \cdot I \cdot (SCM_{it} > \gamma_2) \\ + \sum \eta_i \cdot Control_{it} + \varepsilon_{it} \tag{8}$$

$$\ln CP_{it} = \mu_i + \xi_1 \cdot \ln CM_{it} \cdot I \cdot (D\_Growth_{it} \leq \gamma_1) + \xi_2 \cdot \ln CM_{it} \cdot I \cdot \\ (\gamma_1 < D\_Growth_{it} \leq \gamma_2) + \xi_3 \cdot \ln CM_{it} \cdot I \cdot (D\_Growth_{it} > \gamma_2) \\ + \sum \eta_i \cdot Control_{it} + \varepsilon_{it} \tag{9}$$

$$\ln CP_{it} = \mu_i + \xi_1 \cdot CM_{it} \cdot I \cdot (\ln DSC_{it} \leq \gamma_1) + \xi_2 \cdot CM_{it} \cdot I \cdot \\ (\gamma_1 < \ln DSC_{it} \leq \gamma_2) + \xi_3 \cdot CM_{it} \cdot I \cdot (\ln DSC_{it} > \gamma_2) \\ + \sum \eta_i \cdot Control_{it} + \varepsilon_{it} \tag{10}$$

To test Hypothesis 3: There may be a non-linear relationship between supply chain management and corporate performance. We have developed the following two models using corporate management (D_CM) and solvency (D_DSC) as threshold variables. Models 8 and 9 are shown in Equations (17) and (18).

$$\ln CP_{it} = \mu_i + \xi_1 \cdot \ln SCM_{it} \cdot I \cdot (D\_CM_{it} \leq \gamma_1) + \xi_2 \cdot \ln SCM_{it} \cdot I \cdot$$
$$(\gamma_1 < D\_CM_{it} \leq \gamma_2) + \xi_3 \cdot \ln SCM_{it} \cdot I \cdot (D\_CM_{it} > \gamma_2) \qquad (11)$$
$$+ \sum \eta_i \cdot \text{Control}_{it} + \varepsilon_{it}$$

$$\ln CP_{it} = \mu_i + \xi_1 \cdot \ln SCM_{it} \cdot I \cdot (D\_DSC_{it} \leq \gamma_1) + \xi_2 \cdot \ln SCM_{it} \cdot I \cdot$$
$$(\gamma_1 < D\_DSC_{it} \leq \gamma_2) + \xi_3 \cdot \ln SCM_{it} \cdot I \cdot (D\_DSC_{it} > \gamma_2) \qquad (12)$$
$$+ \sum \eta_i \cdot Control_{it} + \varepsilon_{it}$$

### 4.2.2. Model Results

From Table 3 and Figure 4, we can conclude that Model 1 passes the single threshold test and is significant at the 10% level. This indicates that the threshold variable corporate management (D_CM) significantly affects R&D innovation (lnR&D) while indirectly affecting corporate performance. H1—that there is a non-linear and U-shaped relationship between R&D innovation and corporate performance—was further tested. When the value of corporate management is less than 0.003, increasing R&D investment does not directly increase corporate performance, but when the value of corporate management is more significant than 0.003, continuing to increase R&D investment will cause a rapid increase in corporate performance.

From Table 3 and Figure 4, we can conclude that Model 2 passes the single threshold test and is significant at the 1% level. This indicates that the threshold variable, supply chain management (lnSCM), has a significant threshold effect on R&D innovation (lnR&D), further verifying H1, which states that there is a non-linear relationship between R&D innovation and corporate performance, and a U-shaped relationship. When the value of SCM is less than −7.88, increasing R&D investment will not directly increase corporate performance, but when the value of SCM is greater than −7.88, continuing to increase R&D investment will cause a rapid increase in corporate performance.

From Table 3 and Figure 4, we can conclude that Model 3 passes the single threshold test and is significant at the 1% level. This indicates that the threshold variable D_Growth has a significant threshold effect on R&D innovation (lnR&D), further validating H1—that there is a non-linear and U-shaped relationship between R&D innovation and corporate performance. When the value of growth capability is less than 0, increasing R&D investment will not directly increase corporate performance, but when the value of growth capability is greater than 0, continuing to increase R&D investment will cause a rapid increase in corporate performance.

From Table 3 and Figure 4, we can conclude that although Model 4 passes the double threshold test, both are significant at the 10% level. However, the two thresholds differ by only one ten-thousandth of a percentage point ($\gamma 1 = -0.0001$ and $\gamma 2 = 0$). Therefore, we can approximate this as a single-threshold model. The threshold variable solvency (D_DSC) has a significant threshold effect on R&D innovation (lnR&D), further proving H1. When the value of solvency is less than 0, increasing R&D investment will not directly increase corporate performance, but when the value of solvency is greater than 0, continuing to increase R&D investment will cause a rapid increase in corporate performance.

Finally, from the results of Models 1–4, it is possible to affirm H1, which states that there is a non-linear relationship between R&D innovation and corporate performance and that the relationship is U-shaped. This non-linear relationship between R&D innovation and corporate performance in agroforestry differs from that in previous studies in manufacturing [77].

**Table 3.** Test of the threshold regression effect.

| Explained Variable | Explanatory Variables | Threshold Variable | Threshold | F Value | p Value | Critical Value | | | Threshold Value | 95% Confidence Interval |
|---|---|---|---|---|---|---|---|---|---|---|
| | | | | | | 0.1000 | 0.0500 | 0.0100 | | |
| lnCP | lnR&D | D_CM | Single | 12.32 * | 0.0990 | 12.0810 | 16.5120 | 26.6340 | 0.0030 | [−0.0268, 0.0079] |
| | | | Double | 2.6900 | 0.7830 | 14.6870 | 21.0360 | 42.7240 | 0.0020 | [−0.0496, −0.0101] |
| | lnR&D | lnSCM | Single | 25.62 *** | 0.0160 | 15.3250 | 19.5170 | 29.1820 | −7.8800 | [−0.0354, −0.0024] |
| | | | Double | 9.9800 | 0.2970 | 19.0980 | 27.7770 | 44.8460 | −6.6870 | [−0.0255, 0.0059] |
| | lnR&D | D_Growth | Single | 3.4500 | 0.6000 | 14.8000 | 26.9290 | 52.8290 | 0.0000 | [−0.0230, 0.0097] |
| | | | Double | 45.12 *** | 0.0140 | 20.0380 | 27.1040 | 52.5980 | 0.0000 | [−0.0304, 0.0035] |
| | | | Triple | 5.5800 | 0.4310 | 12.8750 | 18.9660 | 31.0550 | 0.0000 | [−0.1530, −0.0921] |
| | lnR&D | D_DSC | Single | 15.28 ** | 0.0640 | 11.8340 | 17.6100 | 43.3630 | −0.0001 | [−0.0290, 0.0046] |
| | | | Double | 19.37 * | 0.0800 | 16.9630 | 23.9270 | 40.1180 | 0.0000 | [−0.0721, −0.0243] |
| | | | Triple | 6.1200 | 0.5070 | 16.7320 | 24.4200 | 50.3080 | 0.0003 | [−0.0246, 0.0096] |
| | lnCM | SCM | Single | 32.99 *** | 0.0150 | 17.0240 | 21.6770 | 44.7710 | 0.0010 | [ 0.0164,0.0811] |
| | | | Double | 8.3300 | 0.3990 | 23.3880 | 32.2870 | 56.5790 | 0.0000 | [−0.0223, 0.0323] |
| | lnCM | D_Growth | Single | 4.1500 | 0.5140 | 13.2660 | 22.7310 | 39.0060 | 0.0000 | [−0.0056, 0.0548] |
| | | | Double | 32.91 ** | 0.0300 | 16.6240 | 23.7430 | 54.8750 | 0.0000 | [−0.0171, 0.0449] |
| | | | Triple | 7.6600 | 0.2550 | 12.9570 | 16.6910 | 28.0610 | 0.0000 | [−0.1087, −0.0228] |
| | CM | lnDSC | Single | 1.7100 | 0.8150 | 12.7900 | 21.6650 | 97.6070 | −6.2470 | [−0.5864, 1.6827] |
| | | | Double | 60.91 *** | 0.0080 | 18.6640 | 28.7170 | 53.7600 | −6.2530 | [−1.0439, 0.8456] |
| | | | Triple | 2.2900 | 0.8310 | 21.5480 | 33.1740 | 64.7730 | −6.4950 | [7.3769, 13.3909] |
| | lnSCM | D_CM | Single | 12.44 ** | 0.0570 | 10.1270 | 13.3990 | 21.1860 | 0.0030 | [0.0146, 0.0661] |
| | | | Double | 2.3700 | 0.8330 | 13.4580 | 21.5900 | 42.3600 | 0.0020 | [−0.0032, 0.0519] |
| | lnSCM | lnDSC | Single | 16.99 ** | 0.0590 | 12.5520 | 18.3660 | 33.2160 | 0.0000 | [0.0105, 0.0615] |
| | | | Double | 8.0000 | 0.2890 | 14.7900 | 20.6070 | 35.6150 | 0.0003 | [0.0155, 0.0666] |

Note: *** indicates significance at the 1% level; ** indicates significance at the 5% level; * indicates significance at the 10% level.

From Table 3 and Figure 5, we can conclude that Model 5 passes the single threshold test and is significant at the 1% level. This indicates that the threshold variable supply chain management (SCM) has a significant threshold effect on corporate management, further supporting H2, which states that there is a non-linear relationship between corporate management and corporate performance—a U-shaped one. When the value of SCM is less than 0.001, increasing corporate management does not directly increases corporate performance, but when the value of SCM is more significant than 0.001, continuing to increase corporate management will cause a rapid increase in corporate performance.

From Table 3 and Figure 5, we can conclude that although Model 6 passes the double threshold test, both thresholds are zero, and therefore, we regard them as a single threshold. This indicates that the threshold variable growth capability has a significant threshold effect on corporate management, further supporting H2, which states that there is a non-linear relationship between corporate management and corporate performance, a U-shaped one. When the value of growth capacity is less than 0, increasing corporate management does not directly increase corporate performance, but when the value of growth capacity is greater than 0, continuing to increase corporate management will cause a rapid increase in corporate performance.

From Table 3 and Figure 5, we can conclude that although Model 7 passes the double threshold test, the p-value of the first threshold is not significant, so we take the second threshold as a single threshold. This indicates that the threshold variable solvency has a significant threshold effect on corporate management, further supporting H2, which states that there is a non-linear relationship between corporate management and corporate performance and that the relationship is U-shaped. When the value of solvency is less than −6.2530, increasing corporate management does not directly increase corporate performance, but when the value of solvency is greater than −6.2530, continuing to increase corporate management will cause a rapid increase in corporate performance.Finally, the results of Models 5 and 7 support H2.

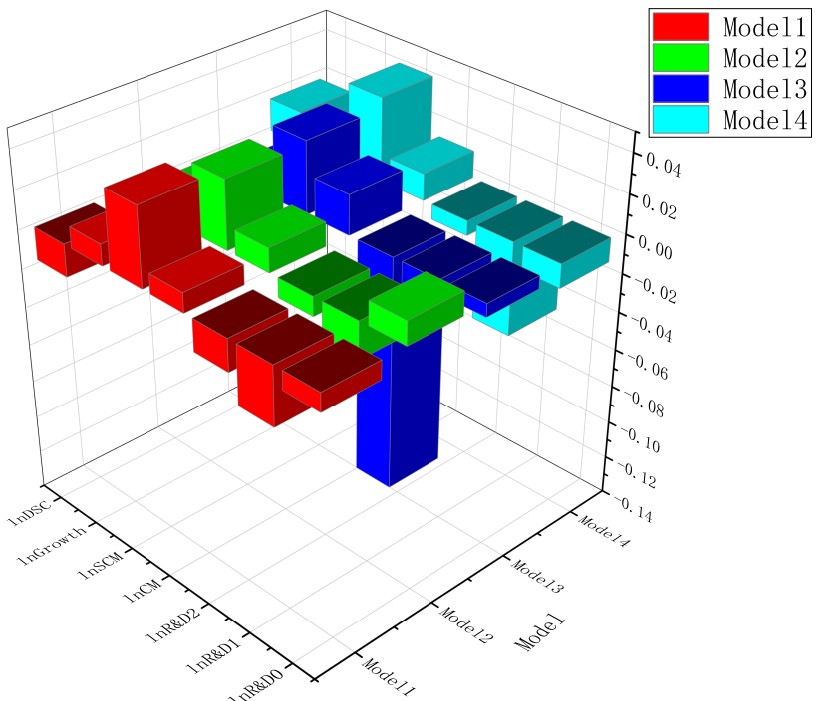

**Figure 4.** Threshold regression results for Models 1–4.

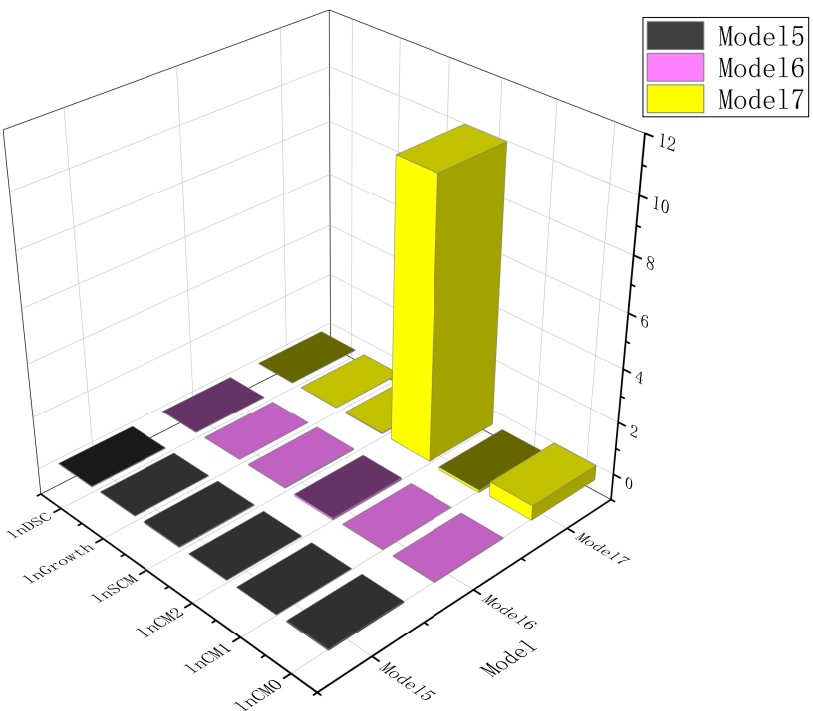

**Figure 5.** Threshold regression results for Models 5–7.

From Table 3 and Figures 6, we can conclude that Model 8 passes the single threshold test and is significant at the 5% level. This indicates that there is a significant threshold effect of the threshold variable company management (D_CM) on supply chain management, further supporting H3, which states that there is a non-linear relationship between supply chain management and corporate performance—an inverted-U-shaped relationship. When the value of CM is less than 0.003, strengthening supply chain management will directly increase corporate performance, but when the value of CM is more significant than 0.003, continuing to strengthen supply chain management will cause corporate performance to decline.

From Table 3 and Figure 6, we can conclude that Model 9 passes the single threshold test and is significant at the 5% level. This indicates that there is a significant threshold effect of the threshold variable solvency (lnDSC) on supply chain management, further supporting H3, which states that there is a non-linear relationship between supply chain management and corporate performance—an inverted-U-shaped relationship. When the value of solvency is less than 0, strengthening supply chain management will directly increase corporate performance, but when the value of solvency is greater than 0, continuing to strengthen supply chain management will cause corporate performance to decline.

Finally, the results of model 8 and model 9 can be used to test H3, which states that there is a non-linear relationship between supply chain management and corporate performance, with an inverted-U-shaped relationship.

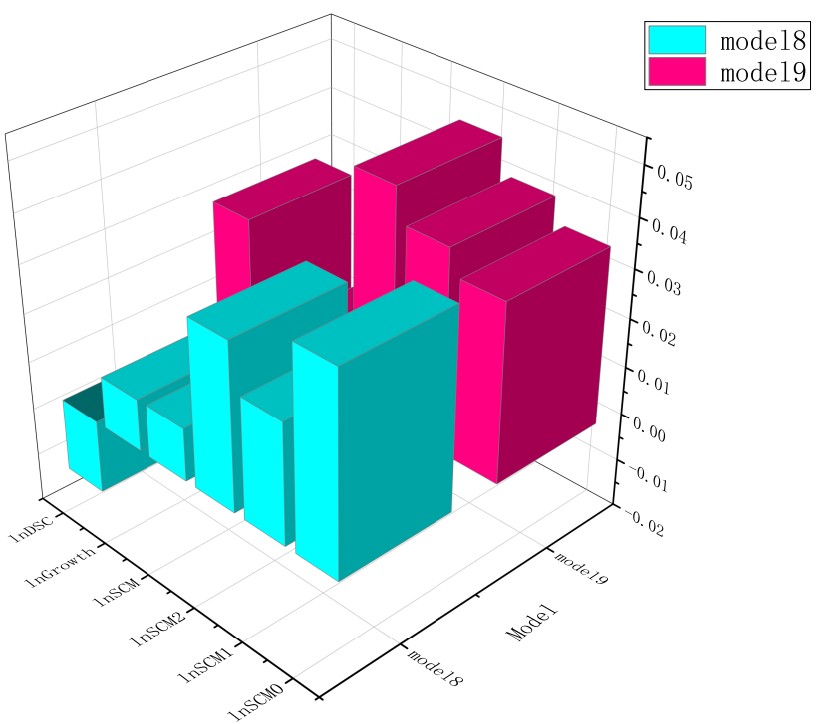

**Figure 6.** Threshold regression results for Models 8–9.

As can be seen in Figures 7 and 8, the LR images of the thresholds all have intersections with the horizontal line, identifying confidence intervals for the threshold values, and based on the data, passing the test of significance.

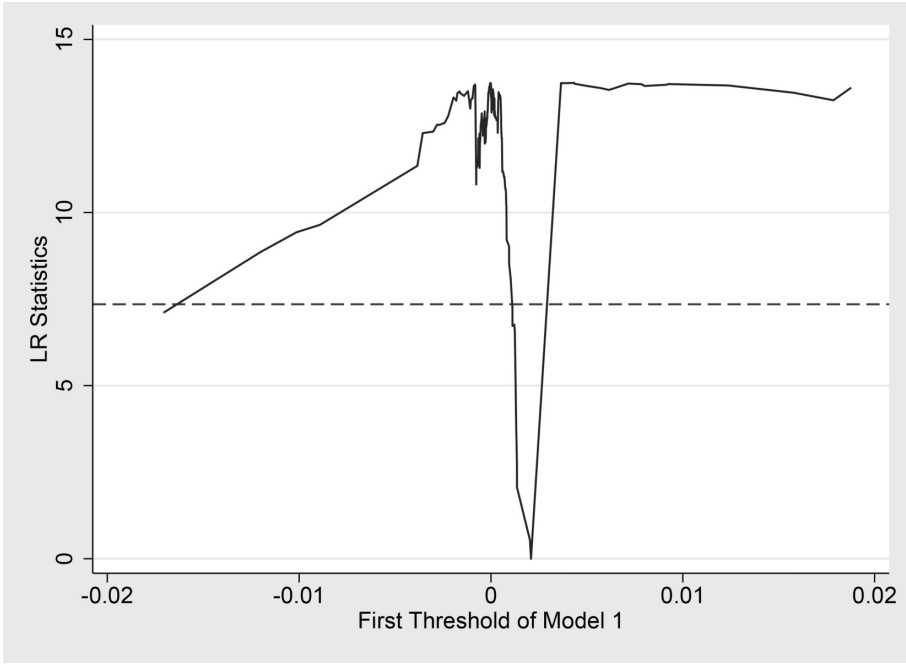

**Figure 7.** LR diagram of Model 1.

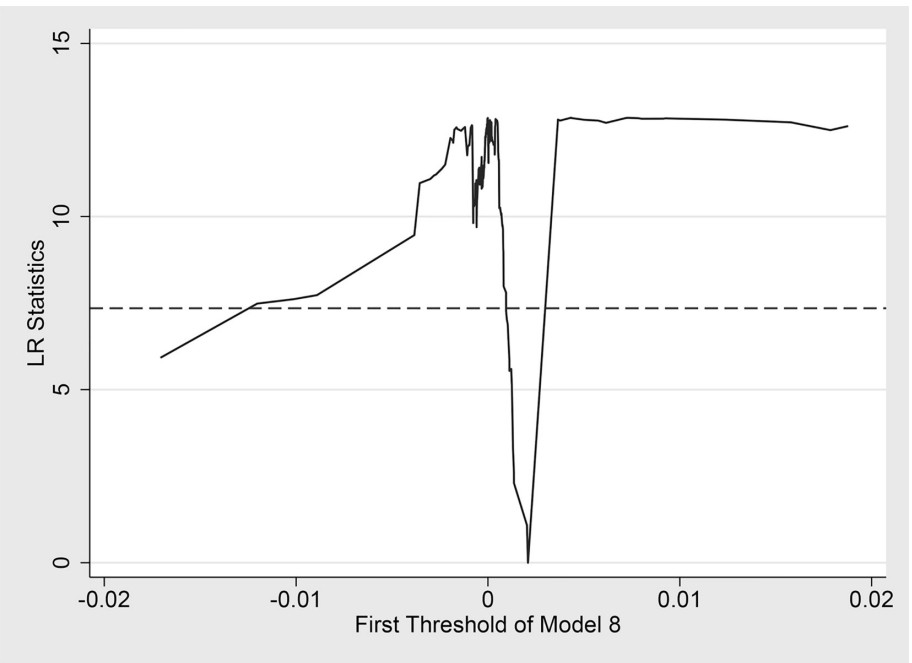

**Figure 8.** LR diagram of Model 8.

## 5. Conclusions and Recommendations

### 5.1. Conclusions

We selected annual panel data on Chinese listed agroforestry companies mainly from the CSMAR database and other sources for the period 2010–2021. We empirically analyzed the panel threshold effect of R&D innovation on the corporate performance of listed agroforestry companies using the entropy weighting method, GMM estimation method, and panel threshold model. The following analytical results and conclusions are finally drawn.

(1) In the development of listed agroforestry companies, there is a non-linear relationship between R&D innovation and corporate performance that is U-shaped, which is different from the conclusions reached by many scholars studying the manufacturing industry. Secondly, corporate management, supply chain management, growth capacity, and debt service capacity all have significant threshold effects on R&D innovation, thereby indirectly affecting the non-linear relationship between R&D innovation and corporate performance.

(2) In the development of listed agroforestry companies, there is a non-linear relationship between corporate management and corporate performance that is U-shaped, which is different from what many scholars have concluded. Supply chain management, growth capacity, and debt service capacity all have significant threshold effects on corporate management, thereby indirectly affecting the non-linear relationship between corporate management and corporate performance.

(3) In the development of listed agroforestry companies, there is a non-linear relationship between corporate management and corporate performance that is inverted-U-shaped, which is consistent with the findings of many scholars. The supply chain is a powerful and effective means to unlock finance, open up the industrial chains of upstream and downstream enterprises, and promote the ecological empowerment of the industrial chain. Both corporate governance and debt service capacity have significant threshold effects on supply chain management, thereby indirectly influencing the non-linear relationship between supply chain management and corporate performance.

### 5.2. Recommendations

(1) We found that the non-linear relationship between R&D investment in listed agroforestry companies and corporate performance is U-shaped, which means that listed agroforestry companies have to increase their R&D investment more than other types

of companies to receive good benefits. This may be related to the characteristics of the agroforestry industry, which is characterized by high investment in R&D, long payback periods, and unresponsiveness to the market. This has led to many listed agroforestry companies in the country being reluctant to invest more in R&D. Therefore, the government should recognize the characteristics of the agroforestry industry and help agroforestry companies to cross the U-shaped inflection point by taking the initiative to reduce taxes or financial subsidies. The government should encourage and support R&D investment in listed agroforestry companies, which, after all, face direct consumer markets and are more aware of market needs. The government should establish a set of laws and regulations to support the development of science and technology innovation in agroforestry listed companies.

(2) We found that the non-linear influence of corporate management on the corporate performance of listed agroforestry companies is also U-shaped, which means that compared with other types of listed companies, listed agroforestry companies have to pay more management costs to obtain higher returns. This industry characteristic may be related to the traditional Chinese "small farmer economy" mentality. Farmers are unwilling to change their long-standing customary farming methods quickly, are less receptive to new management methods, or have a more extended transformation period. These industry characteristics have led to many listed agroforestry companies being unwilling to invest more in management costs and habitually accepting traditional farming methods, which are difficult to change quickly. Therefore, the government should take the initiative to guide the change, relying on the Academy of Agricultural Sciences and other scientific research institutions to actively guide and promote the traditional resource-dependent agroforestry to a technology-intelligent transformation. Changes in agroforestry management should be aimed at effectively increasing farmers' incomes, as the traditional resource-input-based approach to income generation is no longer suitable for the new stage of historical development. Enterprises should not rely too much on government subsidies and policy guidance for a long time but should identify market demand and take proactive action.

(3) We found that the non-linear influence of supply chain management on the enterprise performance of listed agroforestry companies has an inverted "U" shape, which means that compared with other types of listed companies, an appropriate increase in the supply chain management capability of listed agroforestry companies can directly increase enterprise performance. Still, we should pay attention to the reasonable range (however, it should be noted that it is within a reasonable range (around the apex of the "U" shape). This, to some extent, reflects the relatively developed logistics supply chain system in China's agriculture and forestry industry. For the government, while relying on the advantages of the national logistics supply chain system, they can establish a national unified agriculture and forestry market; improve the construction of factories and resources markets; strengthen the quality of services and commodity markets; and unify the rules, standards, and procedures of supervision. Local governments can combine local characteristics to introduce modern management into intensive production, unify sales and purchases, help with sales in a national unified market platform, and reduce the internal consumption of resources in small-scale production. Further, they could make good use of the roles of technological innovation and industrial upgrading, and smooth the national circulation of agricultural and forestry products.

## 6. Research Shortcomings and Perspectives

Although we have made great efforts in collecting data, the data of listed companies in agriculture and forestry given by the CSMAR database, the National Bureau of Statistics, and the Oriental Fortune website are too few, and there are many missing values. For future studies, we will try to collect as much data as possible, or more detailed and comprehensive data through field surveys. Our future research will focus on the mechanism analysis of the causal relationship between R&D innovation, corporate governance, supply chain management, and firm performance.

**Author Contributions:** Y.S.: conceptualization, methodology, funding acquisition, writing—review and editing. H.L.: conceptualization, methodology, data curation, visualization, formal analysis, writing—original draft, writing—review and editing. J.L.: conceptualization, methodology, validation, formal analysis, writing—original draft, writing—review and editing. M.S.: conceptualization, methodology, writing—review and editing. Q.L.: conceptualization, resources, writing—review and editing. All authors have read and agreed to the published version of the manuscript.

**Funding:** Postdoctoral Innovation Program, Chinese Academy of Social Sciences: "Study on the Rural Revitalization of the Characteristic Villages of Ecological Civilization in Wuling Mountain Area" (serial number: IQTE202003).

**Institutional Review Board Statement:** Not applicable.

**Informed Consent Statement:** Not applicable.

**Data Availability Statement:** Data will be made available on request.

**Acknowledgments:** Thanks for the support of funding and NJFU.

**Conflicts of Interest:** The authors declare no conflict of interest.

## Appendix A

*Step* 1: Determine whether there are negative numbers in the input matrix, and if so, renormalize to a non-negative interval.

$$Z_{ij} = \frac{x_{ij} - \min\{x_{1j}, x_{2j} \cdots, x_{ij}\}}{\max\{x_{1j}, x_{2j} \cdots, x_{ij}\} - \min\{x_{1j}, x_{2j} \cdots, x_{ij}\}} \tag{A1}$$

*Step* 2: Calculate the weight of the $i_{th}$ sample under the $j_{th}$ indicator and consider it as the probability used in the relative entropy calculation. Calculate the probability matrix $P_{ij}$.

$$P_{ij} = \frac{Z_{ij}}{\sum_{i=1}^{n} Z_{ij}} \tag{A2}$$

*Step* 3: Calculate the information entropy $e_j$ of each indicator, calculate the information utility value $d_j$, and normalize it to obtain the entropy weight of each indicator.

$$e_j = -\frac{1}{\ln n} \sum_{i=1}^{n} p_{ij} \ln(p_{ij}), (j = 1, 2, \ldots, m) \tag{A3}$$

Among them,

$$k = \frac{1}{\ln n} > 0, e_j \geq 0 \tag{A4}$$

$$d_j = 1 - e_j \tag{A5}$$

*Step* 4: The weights $w_j$ are calculated for each indicator.

$$w_j = \frac{d_j}{\sum d_j} \tag{A6}$$

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
