# Peer review of "Analysis of Factors Influencing the Corporate Performance of Listed Companies in China’s Agriculture and Forestry Sector Based on a Panel Threshold Model"

_sustainability, doi:10.3390/su15020923_

Round 1
Reviewer 1 Report
Dear authors,
Thank you very much for sending your paper to the journal. The topic is interesting; however, the following issues should be fully addressed in the next version:
1- The abstract should be shortened as much as possible, containing the purpose, method, conclusion and originality
2-all variables should have a reference which is employed in the study
3-There are several figures on the paper; some are unnecessary and should be deleted from the paper.
4-The further to the study should be included in the next version
Author Response
Firstly, we are very grateful for your comments. Secondly, please see the attached PDF for a detailed response to your comments.

Reviewer 2 Report
The authors deliberate on the factors influencing the corporate performance of listed agroforestry companies, thus regarding the compound development in agriculture and forestry for a country. Although the area of research is interesting, the authors need to incorporate the following factors to improve this manuscript:
1. Abstract contains a lot of discussion on the results. While the research gap and authors' contribution need more detailing. 2. Keywords include R&D innovation, which does not seem among the mouse relevant in this paper. 3. The main contributions and main innovations are in separate para in Introduction; put those either in bullets or merge them in a single paragraph. 4. The review is hardly complete. They should study more recent articles and enlist some of those in Reference. 5. Is the Sec 3.4 authors' contribution or existing knowledge? If existing, put them in the Appendix. 6. Table 2 should be transposed to improve readability. Similarly, Table 3 can be reoriented. 7. To test hypothesis 2, authors established non-linear relationship between corporate management and corporate performance, without going into details. Please provide references appropriately. Similar concerns are against other hypotheses. 8. A number of typos should be taken care of.Author Response
Firstly, we are very grateful for your comments. Secondly, please see the attached PDF for a detailed response to your comments.

Reviewer 3 Report
please, see attachment.

Author Response

(The authors gave the same response as above.)

Round 2
Reviewer 1 Report
Dear authors,
Thank you very much for sending your revised manuscript. I reviewed the current version and you incorporated my all comments in the current version